# Effect of Flow on the Energy Conversion Characteristics of Multiphase Pumps Based on Energy Transport Theory

Manqi Tang [1,2,*] , Guangtai Shi [1,2,*], Wenjuan Lv [1,2], Xiaodong Peng [1,2] and Zongliu Huang [1,2]

[1] Key 1 Laboratory of Fluid Machinery and Engineering, Xihua University, Chengdu 610039, China; lwj@mail.xhu.edu.cn (W.L.); mypxiaod@126.com (X.P.); hz127@xhu.edu.cn (Z.H.)
[2] Key 2 Laboratory of Fluid and Power Machinery, Xihua University, Ministry of Education, Chengdu 610039, China
[*] Correspondence: tangmanqi7@163.com (M.T.); shiguangtai_1985@126.com (G.S.)

**Abstract:** Multiphase pumps operate under different flow conditions with different work performance. In order to reveal the energy conversion regulations in multiphase pumps under different flows, this paper presents an analysis of the effects of different flows on the pressure propulsion power, Lamb vector dispersion, and vortex enstrophy dissipation in the pressurization unit of a multiphase pump based on energy transport theory. It is found that at different flows, the pressure propulsion power near the impeller inlet decreases sharply, the pressure propulsion power is mainly located in the first half of the impeller near the suction side of the blade, and with the increase in the flow, the pressure propulsion power in the pressurization unit increases gradually, as well as its energy loss, while the Lamb vector dispersion gradually increases and the area of scattering region tends to be narrow under the small impeller tip clearance, while the Lamb vector dispersion region area slowly decreases with the flow rate when the impeller tip clearance is larger. The effect of flow on the vortex enstrophy dissipation in the multiphase pump is mainly located in the middle of the impeller near the blade pressure surface, and as the flow increases, the value of the vortex enstrophy dissipation in the impeller pressurization unit increases accordingly, and the vortex enstrophy dissipation in the first half of the impeller is even more chaotic. The investigation results have significant theoretical meaning for the deep mastery of the energy conversion characteristics in multiphase pumps.

**Keywords:** energy transport theory; flow; multiphase pump; energy conversion characteristics

## 1. Introduction

With the continuous development of the global economy, the requirement for oil and natural gas is gradually increasing, and thus enhancing the high-efficiency exploitation of oil and gas resources has become an important link to consolidate the foundations of resource succession [1–3]. Nevertheless, during the exploration and development of oil and gas resources, the requirement for oil and gas transmission equipment is getting higher and higher, and thus multiphase mixing and transmission technology has come into being against this background [4,5]. The core equipment of multiphase mixing and transmission technology is a multiphase pump (blade type), which has become a focus of research in recent years due to its advantages related to it small size, compact structure, large flow, and operation under a high gas content [6,7]. As the impeller of the multiphase pump is its main work component [8,9], the impeller design's advantages and disadvantages directly determine the properties of the multiphase pump; in order to design an excellent multiphase pump impeller, we must establish the effect of various factors on its energy conversion characteristics.

The energy conversion process could impact the properties of fluid machinery, so research on the energy conversion characteristics of fluid machinery is crucial to improve the energy conversion efficiency of fluid machinery. Liu, M. et al. [10] analyzed the energy characteristics and flow field of multiphase pumps with different viscosities, and the results showed that the increase in viscosity and vane height as well as the decrease in flow velocity

increased the turbulence energy in the pump and decreased the energy conversion efficiency. Quan, H. et al. [11] carried out an energy conversion analysis of the impeller domain of a spiral centrifugal pump based on contour lines, and found that the helical section in the front part of the impeller mainly carries out energy transfer, while the centrifugal section in the back half mainly plays the role of energy conversion, which is more conducive to the energy conversion of solid–liquid two-phase flow when the solid–liquid two-phase conveying is carried out using the spiral centrifugal pump with an increase in the size of the solid particles. Zhang, X.L. et al. [12], through vane pump (including mixed-flow pumps, centrifugal pumps, axial pumps) energy conversion mechanism research, explored the pump input power and the conversion relationship between the output power, resulting in the calculation of the input power change rule expression. Liu, Y. et al. [13] proposed a C-slot method to effectively suppress the leakage flow at the top of the blade of the NACA009 hydrofoil and improve its work performance in the tidal energy conversion process. Miao, S. et al. [14–16] investigated the energy conversion characteristics of hydraulic turbines in pump reversal, concluded that the main energy of the impeller to perform work comes from the pressure energy of the fluid rather than kinetic energy, and that the middle area in front of the impeller is the core area of energy conversion. Zhang, J. et al. [17,18] examined the effects of the gas volume fraction and impeller tip clearance on the work performance and flow pattern of the multiphase pump, which showed that an improvement in both the gas volume fraction and impeller tip clearance would reduce the head and efficiency of the multiphase pump. Shi, G. et al. [19,20] quantitatively catalyzed the energy conversion characteristics in the impeller domain of a multiphase pump, discovering that the fluid medium mainly exchanges energy in the middle region of the impeller and that the improvement of the energy transfer performance of the pressure surface of the impeller has a positive effect on the energy conversion of the impeller. Ji, L. et al. [21] analyzed the effect of different impeller tip clearances on the energy characteristics within a mixed flow pump based on the entropy production theory, and the results showed that an increase in leakage flow may increase the energy loss within the impeller. Jin, Y. et al. [22] investigated the energy conversion characteristics of molten salt pumps in different media based on the energy transport theory, and found that there was no significant change in the energy conversion characteristics of different media when the pump geometry was determined. Zhao, Z. et al. [23] used the LES method to perform high-precision numerical calculations for centrifugal pump stall conditions, by using the energy transport theory to analyze the formation and development of separation bubbles. Liu, Y. et al. [24] found that as the impeller tip clearance rises, the head and efficiency of the mixed flow pump decreases accordingly. Wang, L. et al. [25] studied the work performance of semi-open centrifugal pumps, with the result being that the T-shaped vane can effectively reduce the energy loss during the pump operation and increase its head and efficiency. Ye, D. et al. [26] examined the energy conversion characteristics of reactor coolant pumps and discovered that the hydrostatic energy near the pump inlet continues to grow as the coasting time increases, whereas the area of the worm shell with greater hydrostatic energy decreases.

In summary, the study of energy conversion characteristics in the pump is extremely crucial to improving the energy conversion efficiency of the pump. Many scholars have, in a wide range of research and based on different methods, analyzed the pump energy conversion characteristics of the pump energy conversion process in great depth, revealing the operation process and energy law of the pump. However, there are few studies on the energy conversion characteristics of multiphase pumps based on energy transport theory. Therefore, based on previous research results, this paper will study the energy conversion characteristics of multiphase pumps based on energy transport theory, thoroughly analyze the changes in the pressure propulsion work, Lamb vector dispersion, and vortex enstrophy dissipation in the multiphase pump, and provide a reference for the in-depth mastery of the energy conversion characteristics in multiphase pumps.

## 2. Energy Transport Theory

According to the hydrodynamic theory, the transport equations are as follows:

$$E = \frac{1}{2}|u|^2 \tag{1}$$

$$\rho\frac{DE}{Dt} = \frac{\rho}{2}\frac{d(u^2)}{dt} \tag{2}$$

$$\rho\frac{DE}{Dt} = \rho u\frac{du}{dt} \tag{3}$$

Combining the N–S equations for incompressible fluids:

$$\frac{du}{dt} = f - \frac{1}{\rho}\nabla p + \frac{1}{\rho}\mu\nabla^2 u \tag{4}$$

where $u$ is the velocity vector; $f$ is the external force received per unit volume of fluid, the effects of which are generally not considered in the flow field; $\mu$ is the viscosity coefficient of the fluid; and $\rho$ is the density of the fluid.

Without considering the external forces on the fluid, the energy transport equation is:

$$\rho\frac{DE}{Dt} = -u(\nabla p) + \mu u(\nabla^2 u) \tag{5}$$

Since the velocity dispersion of an incompressible fluid is zero, i.e., $\nabla u = 0$, we can further simplify the second term of the above equation:

$$\mu u(\nabla^2 u) = -\mu u(\nabla \times u) \tag{6}$$

$$\nabla \cdot (u \times \omega) = \omega \cdot (\nabla \times u) - u \cdot (\nabla \times \omega) \tag{7}$$

$$-u \cdot (\nabla \times \omega) = -|\omega|^2 - \nabla \cdot (u \times \omega) \tag{8}$$

Due to the high Reynolds number of the flow field inside the fluid machinery, the effect of turbulence and viscosity in the flow field inside the multiphase pump must be taken into account, so the effective viscosity needs to be selected, and the final energy transport equation is:

$$\mu_e = \mu_d + \mu_v \tag{9}$$

$$\rho\frac{DE}{Dt} = -u(\nabla p) - \mu\nabla \cdot (\omega \times u) - \mu_e|\omega|^2 \tag{10}$$

From the above equation it can be seen that the energy transport equation is divided into three main components: $-u(\nabla p)$, characterized as the work performed by the pressure gradient in the direction of flow; $-\mu\nabla \cdot (\omega \times u)$, characterized as the action of vorticity and velocity on a viscous fluid, referred to as the Lamb vector dispersion of the viscous fluid; and $-\mu_e|\omega|^2$, characterized as enstrophy dissipation in the flow field due to viscosity and vorticity, referred to as the vortex enstrophy dissipation, while $\mu_e$ is the effective viscosity, $\mu_d$ is the power viscosity, and $\mu_v$ is the turbulent viscosity.

## 3. Research Target

This paper takes the self-developed single-stage spiral axial multiphase pump as the research object, and its main performance parameters are shown in Table 1.

**Table 1.** Main performance parameters of multiphase pump.

| Geometric Parameter | | Work Unit | Numerical Value |
|---|---|---|---|
| Design flow | $Q$ | m$^3$/h | 100 |
| Design speed | $N$ | rpm | 3600 |
| Number of impeller blades | $Z_1$ | (-) | 3 |
| Number of diffuser blades | $Z_2$ | (-) | 11 |
| External diameter | $D$ | mm | 161 |
| Impeller inlet angle | $\alpha_h/\alpha_s$ | ° | 9.05/6 |
| Impeller outlet angle | $\beta_h/\beta_s$ | ° | 27.05/24 |
| Inlet angle of diffuser | $\alpha_h/\alpha_s$ | ° | 0 |
| Outlet angle of diffuser | $\beta_h/\beta_s$ | ° | 35 |

Bladegen 18.1 software is used to model the impeller domain and the diffuser domain. In order to ensure the stability of the flow in the multiphase pump, the upstream and downstream are extended to a certain extent, which ensures the accuracy of the numerical simulation, and the multiphase pump model is shown in Figure 1.

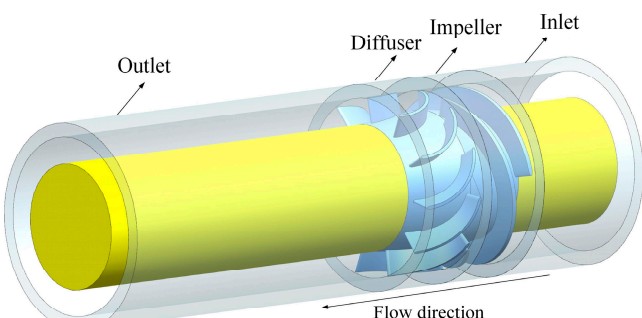

**Figure 1.** Numerical calculation model.

## 4. Mesh Division and Boundary Condition Setting

In this paper, the four parts of the inlet extension section, impeller, diffuser, and outlet extension section of the computational domain are divided by a hexahedral structured mesh, while the O topology is used to control the boundary layer around the blade, and the mesh in the tip clearance is encrypted; the specific mesh is shown in Figure 2.

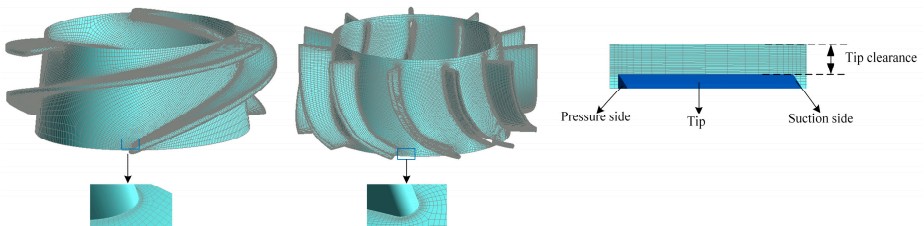

**Figure 2.** The mesh of computational domain.

Four sets of meshes were selected for mesh-independence validation under pure water conditions at a rated flow of 100 m$^3$/h. From Table 2, it can be seen that as the number of meshes increases, the relative head and relative efficiency change within 1%. Considering the efficiency and accuracy of the computational resources, the third set of mesh numbers is selected for the computational study.

**Table 2.** Mesh independence verification.

| Parameters | Mesh 1 | Mesh 2 | Mesh 3 | Mesh 4 |
|---|---|---|---|---|
| Mesh number | 2,246,540 | 2,627,229 | 3,093,387 | 4,054,644 |
| Relative head | 1 | 0.9986 | 1.0023 | 1.0104 |
| Relative efficiency | 1 | 1.0052 | 1.0021 | 0.9976 |

In this paper, the Reynolds time-averaged N–S equations are discretized based on the finite volume method, and the ANSYS CFX 18.1 software is used to simulate the flow model in the multiphase pump under different working conditions. The medium is selected as a gas–liquid two-phase medium (pure water and 25 °C air), the liquid-phase turbulence model is selected as SST, and the gas-phase turbulence model is selected as the dispersed phase zero equation. The boundary conditions are the velocity inlet and pressure outlet, the diameter of the air bubbles is set as 0.1 mm, and the SIMPLE algorithm is used to solve for the pressure and velocity at the same time. In addition, the dynamic-static interface is modeled by the "Frozen rotor", and the stationary domain is treated using the

"GGI" method. The convergence method is RMS with a convergence accuracy of $10^{-5}$, the rotational speed is set to 3000 rpm, the wall is set to be fixed without slip, and the "Scalable Function" is used in the near-wall region.

## 5. Experimental Procedure

To verify the accuracy of the numerical simulation results, a multiphase pump experimental platform was built independently, as shown in Figure 3. Figure 4 shows the experimental system diagram of the multiphase pump, which was mainly composed of four parts: a gas delivery system, a liquid delivery system, a gas–liquid mixing system, and a multiphase pump test system. To visualize the multiphase pump pressurization unit for the test, the shell material used was plexiglass. To ensure the accuracy and reliability of the numerical simulation, the multiphase pump with the impeller tip clearance of 1.0 mm was tested for external characteristics at a rotational speed of 3000 rpm under pure water conditions (flow of 60–120 m$^3$/h) and gas–liquid two-phase condition (GVF = 5%), while the experimental results were summarized with the numerical simulation results and the external characteristic curve was drawn, as shown in Figures 5 and 6. Figure simulation and tests of the head and efficiency were performed, and the output power error was less than 5%, which is unavoidable, so this verifies that the numerical simulation calculation method selected in this paper is reliable.

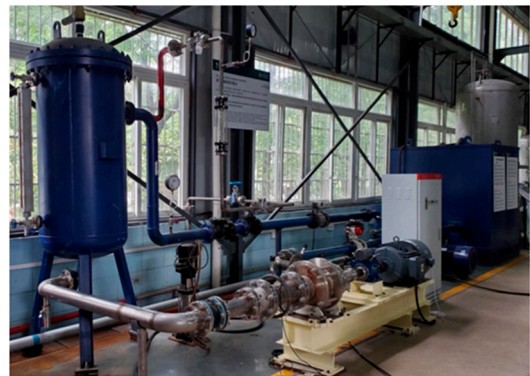

**Figure 3.** Experimental platform for the multiphase pump.

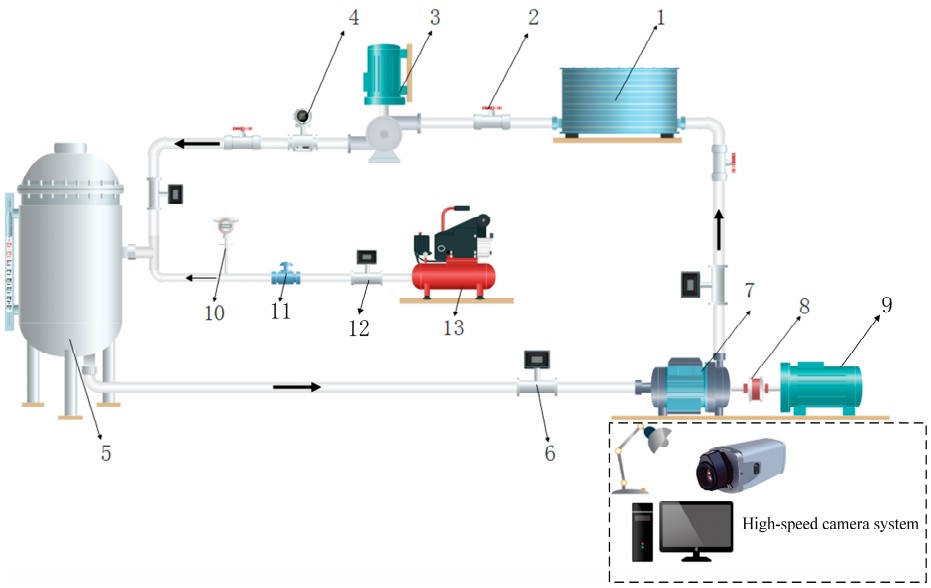

**Figure 4.** Experimental system diagram of multiphase pump. 1—Water tank; 2—regulation valve; 3—booster pump; 4—flowmeter; 5—gas–liquid mixing tank; 6—pressure transducer; 7—multiphase pump; 8—torquemeter; 9—motor; 10—gas flowmeter; 11—regulation valve; 12—barometer; 13—air compressor.

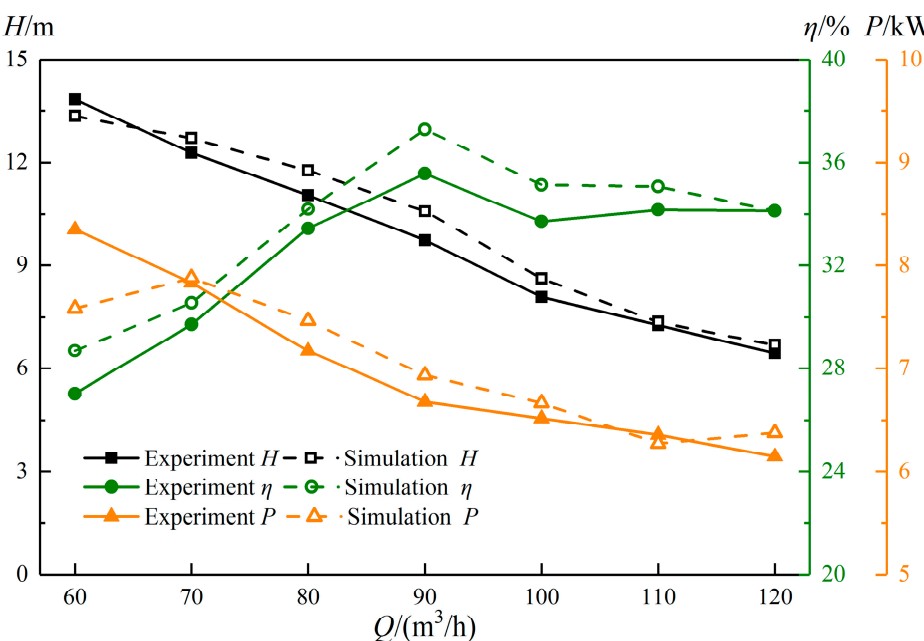

**Figure 5.** Comparison of experimental and numerical simulation under pure water conditions.

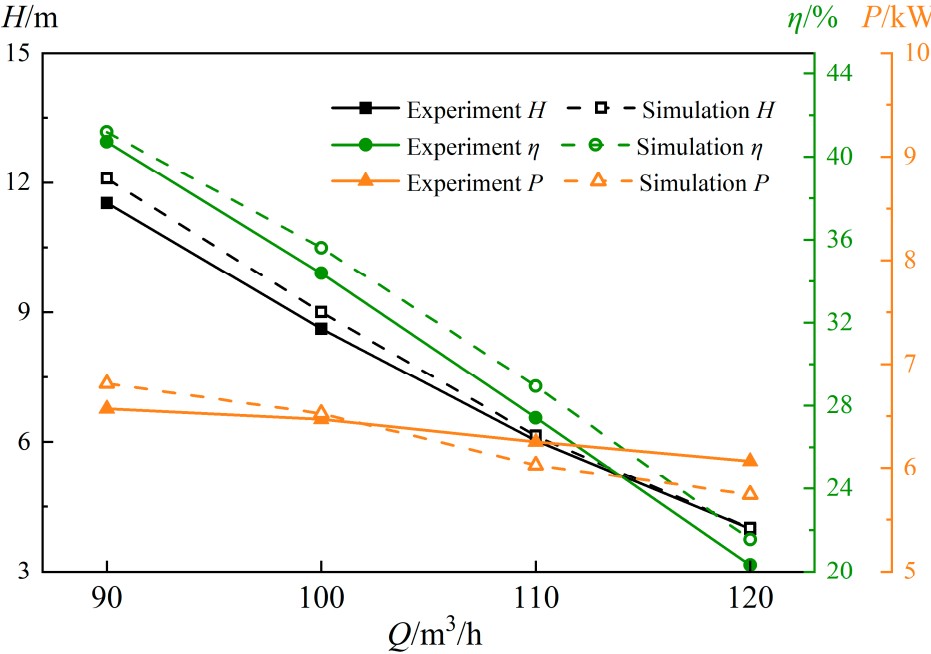

**Figure 6.** Comparison of experimental and numerical simulation under gas–liquid two-phase conditions.

## 6. Analysis of Results

### 6.1. Effect of Flow on the Pressure Propulsion Power within a Pressurization Unit

Figure 7 displays the effect of flow on the work performed by the pressure gradient in the multiphase pump. As can be seen from Figure 6, under different flow conditions, the pressure propulsion power decreases gradually with the increase in the impeller tip clearance, but the overall curve changes are more consistent, showing that the pressure propulsion power at the impeller inlet produces a reduction in the sudden change followed by a rise in the trend, until the streamwise of about 0.5 at the position of the pressure propulsion power changes to a gentle decline. With the increase in flow, the amplitude of pressure propulsion power change in the impeller inlet increases, and the peak pressure propulsion power shows an increase with the increase in flow. Overall, due to the direct collision of the leading edge of the blade

and the fluid with the front end of the inverse pressure gradient, causing a large energy loss, the pressure propulsion power at the impeller inlet appears to drop sharply. With the increase in the flow of the pressure, the gradient falls more sharply, followed by a gradual increase along the axial direction until the pressure propulsion power of the impeller's back half is gradually reduced, which indicates that the pressure propulsion power is mainly located in the first half of the impeller and that the flow has a greater impact. At the same time, because of the effect of dynamic and static interference between the impeller and the diffuser, there are energy fluctuations at the impeller outlet, but they are less affected.

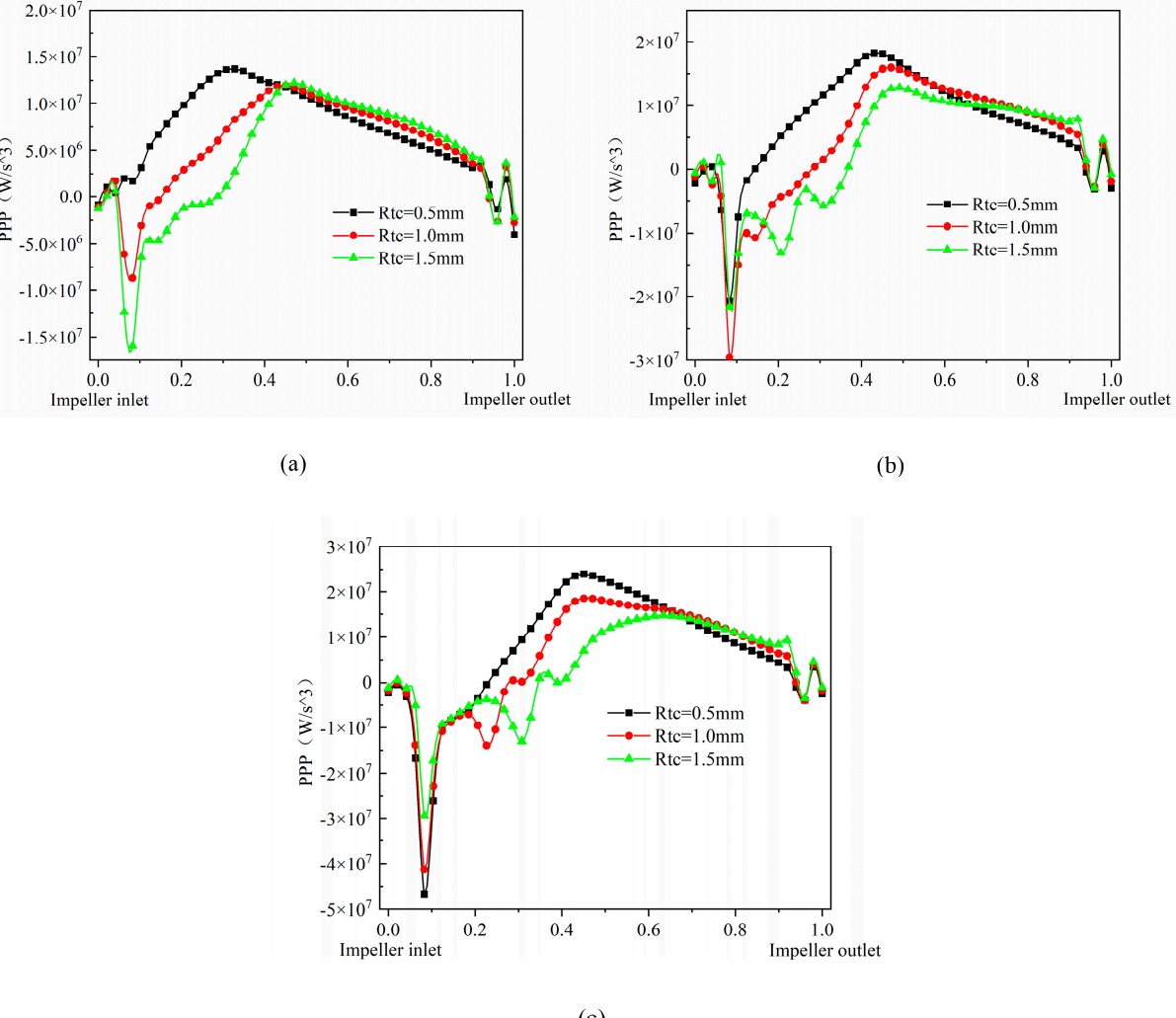

**Figure 7.** 0.8Q (**a**), 1.0Q (**b**), 1.2Q (**c**). Pressure propulsion power distribution in impeller under different flow conditions.

Figure 8 shows the effect of flow on the distribution of pressure propulsion power in the pressurization unit of the multiphase pump at 0.5 times the blade span. From Figure 7, it can be seen that under the 0.8Q flow conditions, the area of the region with larger pressure propulsion power is located in the front half of the blade pressure's front section near the suction surface of the blade, and the area of the high-energy region increases with the increase in the impeller tip clearance. In the 1.0Q flow conditions, the area of the pressure propulsion power region in the pressurization unit is significantly enlarged compared with the former, but the trend of the high energy change is opposite to that of the former. Under the 1.2Q flow conditions, the pressure propulsion power area of the pressurization unit is as high as one-half of the blade passage, which is more variable compared to the previous two flow conditions, and the suppression of the pressure propulsion power in the passage is more obvious with

the increase in the tip clearance. In general, the range of pressure propulsion power in the pressurization unit grows with the increase in flow, and due to the influence of static and dynamic interference between the impeller and the diffuser, the law of pressure gradient change is more chaotic, so the phenomenon of positive and negative pressure propulsion power values will be present alternately.

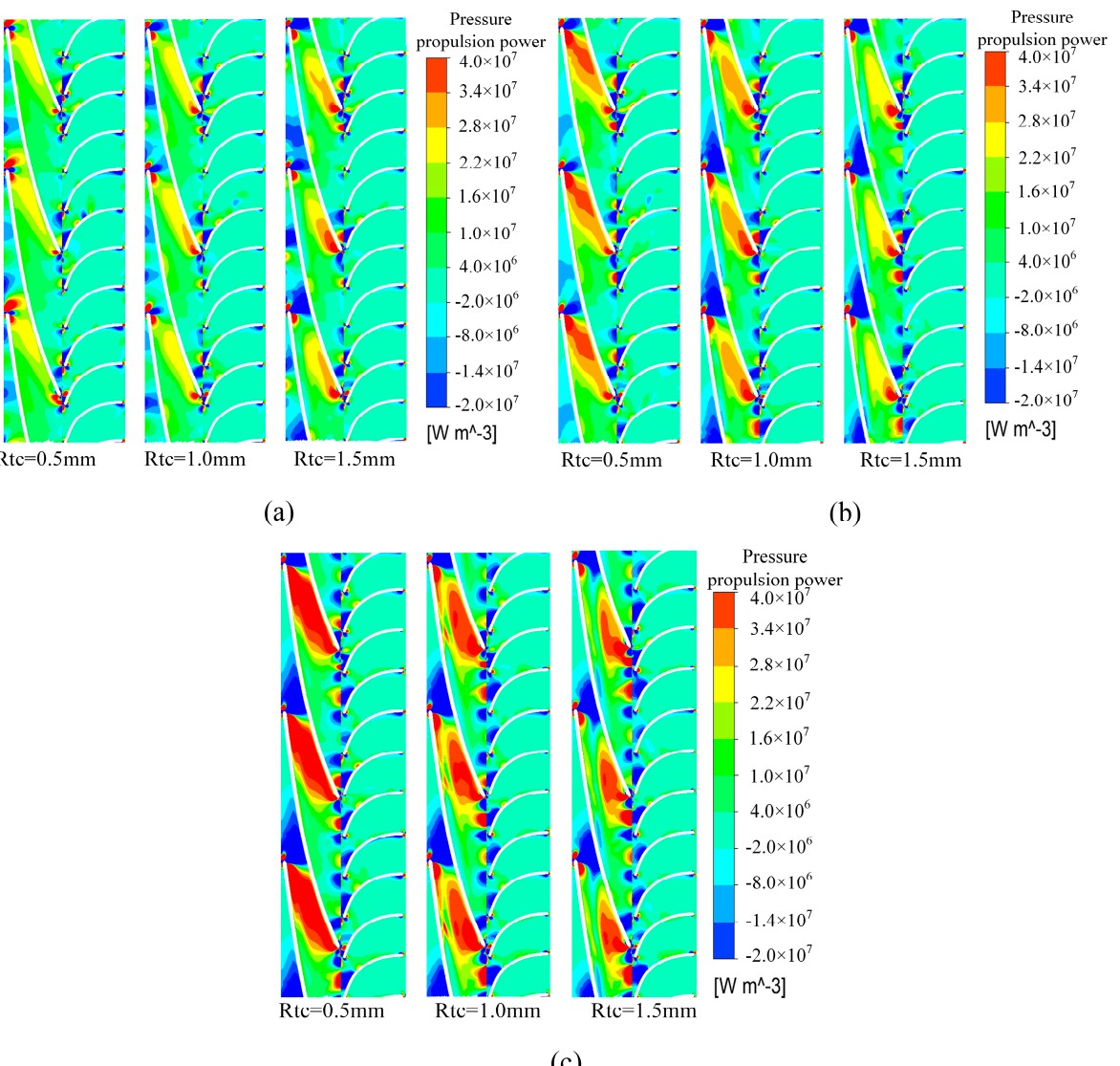

**Figure 8.** 0.8Q (**a**), 1.0Q (**b**), 1.2Q (**c**). Distribution of pressure propulsion power in the pressurization unit with different flow conditions.

### 6.2. Effect of Flow on the Lamb Vector Dispersion with a Pressurization Unit

Figure 9 shows the distribution of the Lamb vector dispersion in the impeller of the multiphase pump under different flows. Lamb vector dispersion is a characteristic quantity that characterizes the vortex force in the flow field, which is very important for the construction of the flow field and can be used to describe the momentum exchange between high- and low-momentum fluids. Its positive and negative values represent the stretching deformation of the momentum transport and vortex motion, respectively, and its positive and negative alternation characterizes the change in the form of the momentum transport of the fluid medium. As can be seen from Figure 8, in the first two of the different flow conditions, the impeller momentum transport is mainly dominated by the vortex motion, and after the axial coefficient is roughly at the position of 0.6, its momentum transport is dominated by the

stretching motion of the vortex. Under the 0.8Q flow conditions, with the increase in the impeller tip clearance, the larger the Lamb vector dispersion value of the front part of the impeller inlet is, the smaller the Lamb vector dispersion value of the back part of the impeller outlet is. At 1.0Q flow, the Lamb vector dispersion is increased by the flow, presenting a more chaotic Lamb vector dispersion value, displaying alternating positive and negative values, and the change in its Lamb vector dispersion is more prominent with the increase in the impeller tip clearance. Under the 1.2Q flow conditions, the impeller momentum transport is dominated by the vortex motion, and the minimum value of Lamb vector dispersion in the impeller gradually reduces with the increase in the impeller tip clearance. Due to the change in flow, the Lamb vector dispersion distribution shows alternating positive and negative states, and the fluctuation is extremely strong at the impeller inlet. In addition, the vicinity of the impeller outlet is affected by dynamic and static interference, which also shows alternating positive and negative changes in the Lamb vector dispersion.

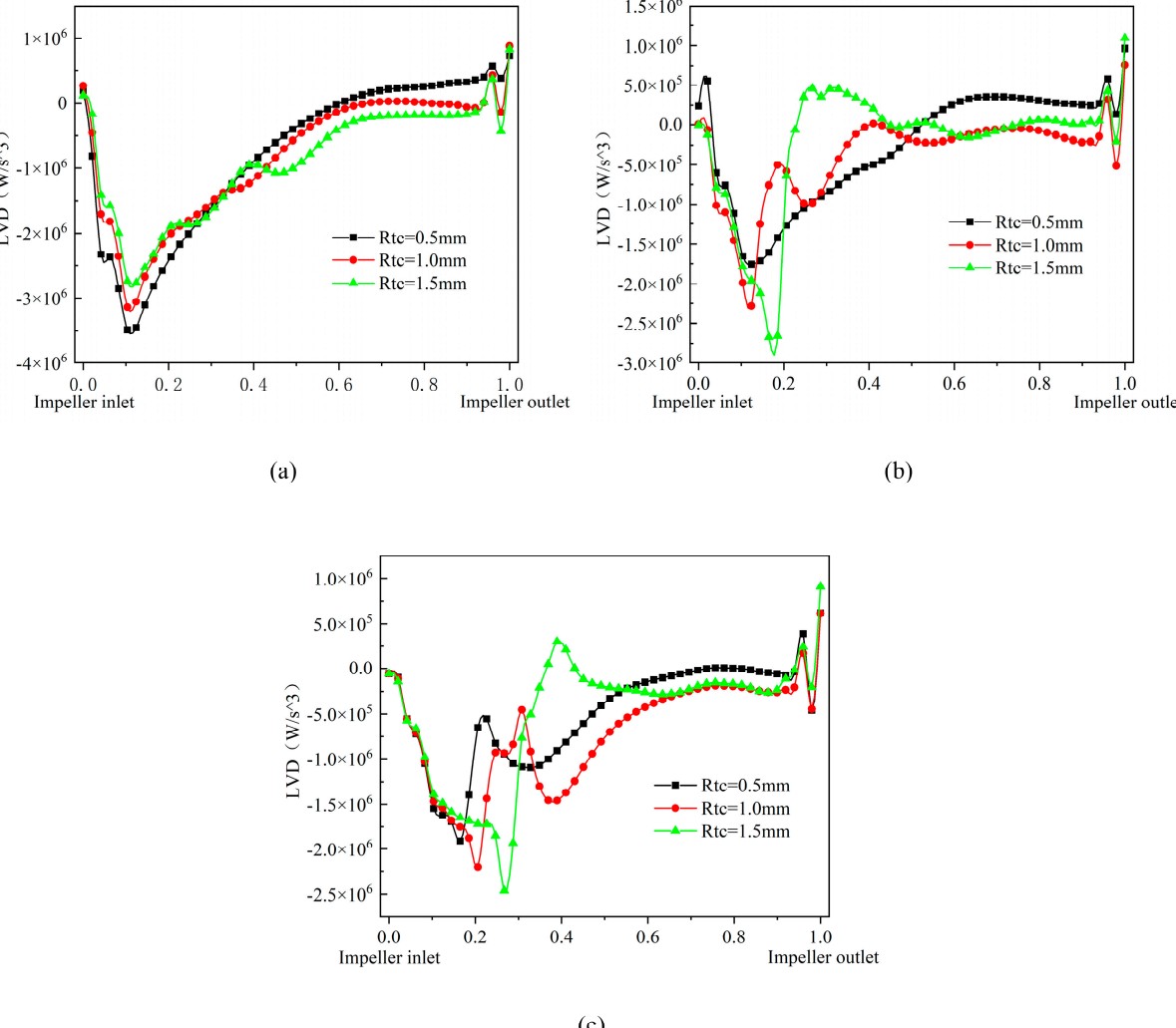

**Figure 9.** 0.8Q (**a**), 1.0Q (**b**), 1.2Q (**c**). Distribution of Lamb vector dispersion in impeller with different flow conditions.

Figure 10 reflects the effect of different flow conditions on the Lamb vector dispersion distribution of the multiphase pump at 0.5 times the blade span. In general, the Lamb vector dispersion is negative on the blade surface and positive in the middle of the flow passage, which is beneficial for preventing the roll-up of the shear layer vortex and thus reducing the instability of the shear layer, in addition to a certain amount of positive and negative values alternating within the diffuser. As can be seen in Figure 9, under the 0.8Q flow conditions,

the region with high Lamb vector dispersion is mainly located in the middle of the flow passage near the pressure surface of the blade, and the area of this region stretches backward and increases with the increase in the impeller tip clearance, gradually forming a long and narrow region of positive Lamb vector dispersion A (as shown in the label of Figure 9a), which enhances the energy exchange between high-and low-momentum fluids in the impeller. At 1.0Q flow, the middle of the flow passage under 0.5 mm impeller tip clearance has a teardrop-shaped high-Lamb vector dispersion region, and the area of this region increases significantly with the increase in the impeller tip clearance. At 1.2Q flow, the high Lamb vector dispersion in the middle of the runner under the small impeller tip clearance shows a needle and leaf shape, and the area of the region gradually shrinks with the increase in the impeller tip clearance, while it increases dramatically under the 1.5 mm impeller tip clearance, and the energy growth law of the vortex pulling deformation at the impeller tip clearance is similar to the former. In summary, it can be seen that as the flow increases, the Lamb vector dispersion under 0.5 mm impeller tip clearance gradually increases and the area of the scattering region tends to be narrow and long, while the area of the Lamb vector dispersion region under 1.0 mm impeller tip clearance and 1.5 mm impeller tip clearance decreases slowly as the flow increases so that the change in flow has a greater impact on the Lamb vector dispersion region.

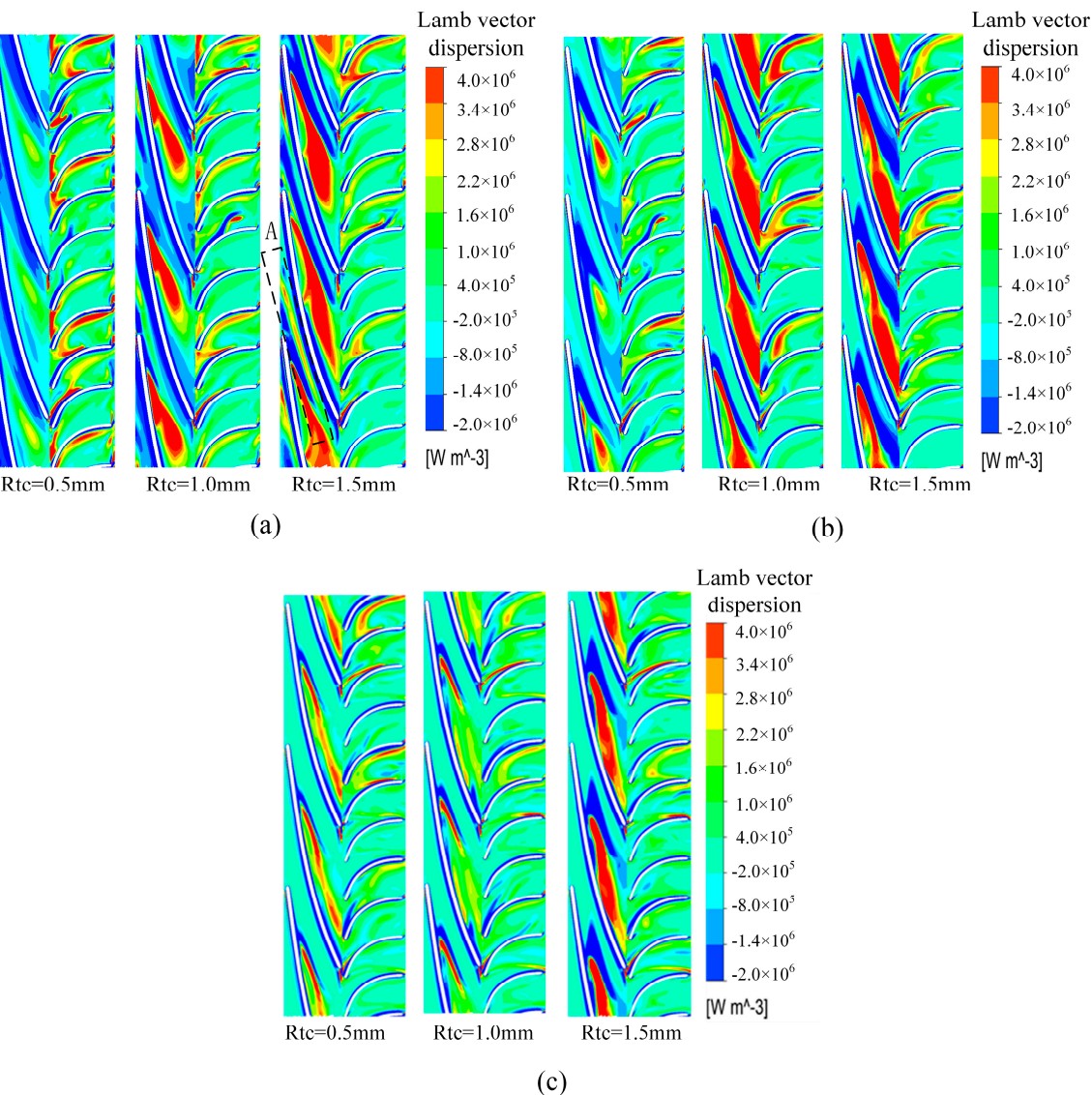

**Figure 10.** 0.8Q (**a**), 1.0Q (**b**), 1.2Q (**c**). Distribution of Lamb vector dispersion in the pressurization unit with different flow conditions.

### 6.3. Effect of Flow on the Vortex Enstrophy Dissipation in the Pressurization Unit

Figure 11 reflects the effect of different flows on the vortex enstrophy dissipation of the multiphase pump. As can be seen from Figure 10, under the condition of 0.8Q flow, the variation rule of vortex enstrophy dissipation is basically similar under different impeller tip clearances, and with the increase in impeller tip clearance, the vortex enstrophy dissipation is also obviously strengthened. At 1.0Q flow, the vortex enstrophy dissipation in the first half of the impeller is more turbulent, the trend of the second half of the impeller is similar to the 0.8Q flow, and the vortex enstrophy dissipation value is greater than the 0.8Q flow. In the 1.2Q flow conditions, the direct collision between the leading edge of the blade and the fluid caused a large amount of energy loss, so the impeller inlet vortex enstrophy dissipation is more chaotic, the trend of the second half of the impeller is similar to the first two flow conditions, and the peak value of the vortex enstrophy dissipation is not very different from the 1.0Q flow conditions. In general, as the flow increases, the vortex enstrophy dissipation in the impeller pressurization unit increases accordingly, and the vortex enstrophy dissipation in the first half of the impeller is more chaotic.

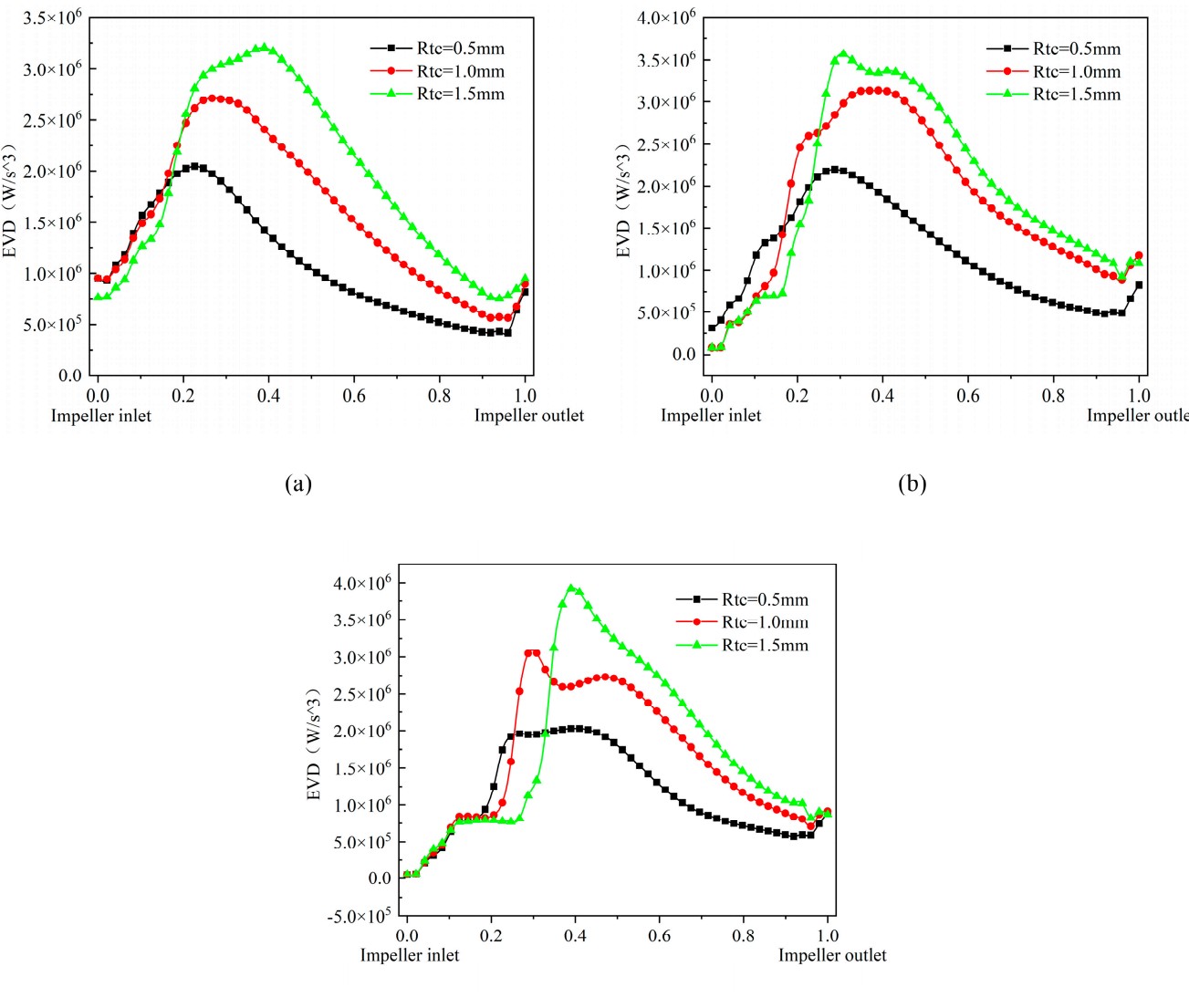

**Figure 11.** 0.8Q (**a**), 1.0Q (**b**), 1.2Q (**c**). Distribution of the vortex enstrophy dissipation in the impeller with different flow conditions.

Figure 12 shows the distribution of vortex enstrophy dissipation in the pressurization unit at 0.5 times the blade span for different flow conditions. As can be seen from Figure 11, the high vortex enstrophy dissipation region in the impeller at 0.5 times the blade span is mainly located in the middle of the blade passage near the impeller pressure surface. At 0.8Q flow, the increase in the impeller tip clearance enlarges the area of the region with higher vortex enstrophy dissipation in the impeller. In the 1.0Q flow conditions, with the increase in the tip clearance of the impeller, the vortex enstrophy dissipation law is more chaotic, and due to the increase in the flow, the overall pressurization unit vortex enstrophy dissipation is relatively slow to increase. In the 1.2Q flow conditions, the vortex enstrophy dissipation in the pressurization unit is significantly smaller compared to the first two conditions, the tip clearance on the vortex enstrophy dissipation in the pressurization unit is similar to the influence of small flow conditions, and only in the large-tip-clearance impeller passage in the region of obvious high enstrophy dissipation. It can be seen that under different flow conditions, the change in flow has a significant effect on the vortex enstrophy dissipation.

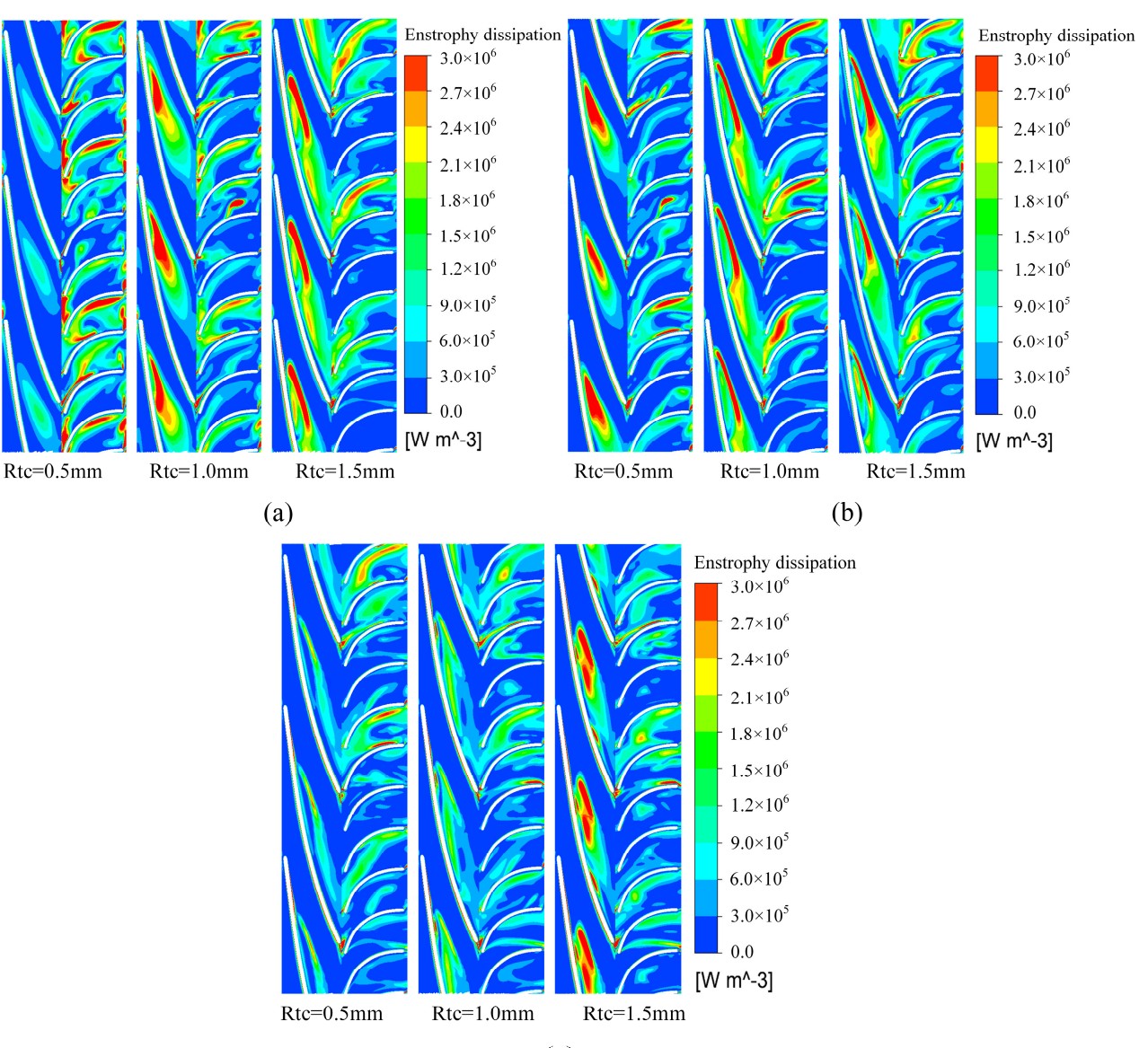

**Figure 12.** 0.8Q (**a**), 1.0Q (**b**), 1.2Q (**c**). Distribution of the vortex enstrophy dissipation in the pressurization unit with different flow conditions.

## 7. Conclusions

In this paper, energy transport theory is applied to analyze the influence of the tip clearance on the pressure propulsion power, Lamb vector dispersion, and vortex enstrophy dissipation in the pressurization unit of a multiphase pump under different flows, and to further investigate the influence of the tip clearance on the energy conversion of the multiphase pump. The main conclusions are as follows:

(1) At different flows, the pressure propulsion power near the impeller inlet decreases sharply, and the pressure propulsion power is mainly located in the first half of the impeller near the suction surface of the blades, while the pressure propulsion power performance improves with the reduction in the impeller tip clearance. With the increase in flow, the pressure propulsion power in the pressurization unit gradually increases, and its energy loss also increases.

(2) Under different flow conditions, the Lamb vector dispersion is negative on the blade surface and positive in the middle of the flow passage, which is beneficial for preventing the roll-up of the shear layer vortex and thus reducing the instability of the shear layer. With the increase in flow, the Lamb vector dispersion gradually increases and the area of the scattering region tends to be narrow with a small impeller tip clearance, while the area of the Lamb vector dispersion region decreases slowly with the increase in flow when the impeller tip clearance is large, i.e., the change in flow has a greater influence on the Lamb vector dispersion region.

(3) The effect of flow on the vortex enstrophy dissipation in the multiphase pump is mainly located in the middle of the impeller near the impeller pressure surface, and with increasing flow, the value of vortex enstrophy dissipation in the impeller pressurization unit increases accordingly, and the vortex enstrophy dissipation in the first half of the impeller is even more chaotic. That is to say, at different flows, the flow change has a significant effect on the vortex enstrophy dissipation.

**Author Contributions:** Conceptualization, methodology, software, writing—original draft, writing—review and editing, M.T.; investigation, software, writing—original draft, G.S.; writing—review and editing; investigation, software, conceptualization, W.L.; writing—review and editing, supervision, X.P.; supervision, validation, Z.H. All authors have read and agreed to the published version of the manuscript.

**Funding:** This research was funded by the Central Leading Place Scientific and Technological Development Funds for Surface Project (2021ZYD0038); Major science and technology project of Sichuan Province (22JBGS0009).

**Data Availability Statement:** The study did not report any data.

**Acknowledgments:** This work was supported by the Central Leading Place Scientific and Technological Development Funds for Surface Project (2021ZYD0038); Major science and technology project of Sichuan Province (22JBGS0009). We are grateful for the support that enabled us to complete this work.

**Conflicts of Interest:** The authors declare no conflict of interest.

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
