# Peer review of "Effect of Flow on the Energy Conversion Characteristics of Multiphase Pumps Based on Energy Transport Theory"

_water, doi:10.3390/w15234188_

Round 1
Reviewer 1 Report
Comments and Suggestions for Authors
Based on the energy transport theory, the article analyzes the effects of different flows on the pressure propulsion power, Lamb vector dispersion and vortex enstrophy dissipation in the pressurization unit of the multiphase pump. The article has a clear overall structural hierarchy, but there are still some minor problems, and the following suggestions are given:
1. The author seems to use inaccurate terminology throughout, such as "vortex enstrophy dissipation", "pressurization", etc., and it is advisable to verify its accuracy.
2. In lines 11-12, the author's use of "different flow" is suggested to be modified to "different flows", and this comment applies to the entire paper.
3. In line 41, the author's use of "designed" should be changed to "design", and the author is advised to carefully check the accuracy of the tense, a comment that applies throughout the article.
4. It can be clearly seen that the size of formula (4) is not consistent with the rest of the formulas, so please ask the authors to adjust the formula format and try to ensure the consistency of the formulas throughout the text.
5. The size of the typography in all tables throughout the text has not been adjusted to be consistent, so authors are asked to determine the appropriate size and make changes.
6. In lines 356 and 359, the article "the" is repeated, and it is recommended that the author verify the accuracy of the use throughout the article.
Comments on the Quality of English Language
Minor editing of English language required
Author Response
Thank you very much for your suggestion. I have uploaded the corresponding response along with the revised manuscript and invite you to review it.。

Reviewer 2 Report
Comments and Suggestions for Authors
Manuscript ID: water-2716991
The manuscript investigates the effect of flow on the energy conversion characteristics of a multiphase pump using energy transport theory. The paper discusses the pressure propulsion power, lamb vector dispersion, and vortex enstrophy dissipation under different flow conditions.
Comments
1. What are the optimal flow conditions for maximizing the energy conversion efficiency in multiphase pumps? Can the study provide recommendations for improving pump design and operation in practical applications?
2. How can the findings of this study be used to optimize the design of multiphase pumps, especially with respect to impeller geometry and clearance? Are there specific design modifications that can enhance energy conversion under various flow conditions?
3. How can the energy losses identified in the study be minimized or mitigated, especially in scenarios where the impeller tip clearance is smaller? Are there engineering solutions that can reduce energy losses without compromising other aspects of pump performance?
4. Are there novel flow control strategies that can be employed to manipulate pressure propulsion power, lamb vector dispersion, and vortex enstrophy dissipation in multiphase pumps to achieve specific operational goals or optimize performance for different applications?
5. Does the observed behavior of pressure propulsion power, lamb vector dispersion, and vortex enstrophy dissipation remain consistent when scaling up or down the size of the multiphase pump? How do these characteristics change with varying pump sizes and capacities?
6. How do the findings of this research apply to real-world industrial applications? Can the insights gained from this study be translated into more efficient and reliable multiphase pumping systems in industries such as oil and gas, chemical processing, or wastewater treatment?
7. Can the research findings be utilized to develop energy recovery systems that harness and convert wasted energy, especially in scenarios where high-pressure propulsion power is dissipated? What are the practical implications for energy-saving technologies?
8. How do different multiphase flow regimes (e.g., gas-liquid, solid-liquid) affect the energy conversion characteristics observed in the study? Are there distinct patterns or principles that apply to each type of multiphase flow?
9. Can advanced numerical simulations and computational models be developed further to explore the intricate flow dynamics within multiphase pumps, allowing for a deeper understanding of energy conversion processes and more precise predictions of pump performance under different flow conditions?
10. What are the environmental and economic implications of improving energy conversion in multiphase pumps? How can more efficient pumps reduce energy consumption and environmental impact in various industrial processes?
Comments on the Quality of English Language
Minor editing of English language required
Author Response
Thank you very much for your suggestion. I have uploaded the corresponding response along with the revised manuscript and invite you to review it.

Reviewer 3 Report
Comments and Suggestions for Authors
this work is to seek answering the exact effects of different flow on the pressure propulsion power, lamb vector dispersion and vortex enstrophy dissipation in the pressurisation unit of the multiphase pump based on the energy transport theory.
GC1, the authors seem seeking answers to too many scientific questions, which make the reviewer feels difficult in figuring out which one actually the authors would like to answer in this work. please try to make the primary scientific question clearer.
Gc2, Please revise the English language. it is a scientific report. then scientific English should be used. A native English speaker could be of help in this case. as an example, in line 165, "its shell material selection of organic glass".
gc3, among the majority of the references in the introduction section, it is not very clear that how those references could be directly related to multiphase flow in a pump. it could be nicer if the authors could make this more obvious.
gc4, in the conclusion section, please brief your work first, before drawing out main conclusions.
sc1, line 10 page 1, "the multiphase pump", is which one?...
sc2, line 91 - 92 on page2, i do not think you can reach this conclusion from the literature summarized above.
sc3, in line 165, "its shell material selection of organic glass", why you chose organic glass? seems a unnecessary detail tome.
sc4, line 173, what dou you mean? please rephrase your expression.
sc5, around line 141 in section 4, regarding the mesh-independence validation, since you are carrying out a multiphase study, please justify a bit more on using pure water in the validation process. better with some references.
Comments on the Quality of English Language
Gc2, Please revise the English language. it is a scientific report. then scientific English should be used. A native English speaker could be of help in this case. as an example, in line 165, "its shell material selection of organic glass".
an improvement in the language, could make this work more accessible to the readers.
Author Response

(The authors gave the same response as above.)

Reviewer 4 Report
Comments and Suggestions for Authors
The multiphase pump operates under different flow conditions with different work performance. In order to reveal the energy conversion regulations in the multiphase pump under different flow, the author presents an analysis of the effects of different flow on the pressure propulsion power, lamb vector dispersion and vortex enstrophy dissipation in the pressurisation unit of the multi-phase pump based on the energy transport theory. The investigation results have significant theoretical meanings for the deep mastery of the energy conversion characteristics in the multiphase pump. However, the article still has some problems that need to be modified:
1. In lines 11-12 of the text, "different flow" appears to be ungrammatical, so please carefully review the text and correct it.
2. In the introduction, the author's name is not uniquely expressed; the author is requested to standardize the expression and make corrections.
3. In lines 106-108, the spacing of the mathematical variables used does not fit the article, resulting in the absence of variables, and it is recommended that the author check the entire text and make changes accordingly.
4. It is recommended that Figure 6-11 be re-edited for layout.
5. In lines 315-316, the word "impeller" is repeated, and the author is advised to correct it.
Author Response

(The authors gave the same response as above.)

Round 2
Reviewer 2 Report
Comments and Suggestions for Authors
None